# Band Structure Studies of the *R*_5_Rh_6_Sn_18_ (*R* = Sc, Y, Lu) Quasiskutteridite Superconductors

**DOI:** 10.3390/ma15072451

**Published:** 2022-03-26

**Authors:** Józef Deniszczyk, Andrzej Ślebarski

**Affiliations:** 1Institute of Materials Engineering, University of Silesia in Katowice, 75 Pułku Piechoty 1A, 41-500 Chorzów, Poland; jozef.deniszczyk@us.edu.pl; 2Institute of Low Temperature and Structure Research, Polish Academy of Sciences, Okólna 2, 50-422 Wrocław, Poland

**Keywords:** superconductivity, atomic disorder, electronic band structure, density functional theory

## Abstract

We report on X-ray photoelectron spectroscopy and ab initio electronic structure investigations of the skutterudite-related $R_{5}$Rh${}_{6}$Sn${}_{18}$ superconductors, where *R* = Sc, Y, and Lu. These compounds crystallise with a tetragonal structure (space group I$4_{1}/acd$) and are characterised by a deficiency of *R* atoms in their formula unit ($R_{5-\delta }$Rh${}_{6}$Sn${}_{18}$, δ≪1). Recently, we documented that the vacancies δ and atomic local defects (often induced by doping) are a reason for the enhancement in the superconducting transition temperature $T_{c}$ of these materials, as well as metallic (δ=0) or semimetallic (δ≠0) behaviours in their normal state. Our band structure calculations show the pseudogap at a binding energy of −0.3 eV for the stoichiometric compounds, which can be easily moved towards the Fermi level by vacancies δ. As a result, dychotomic nature in electric transport of $R_{5}$Rh${}_{6}$Sn${}_{18}$ (metallic or semimetallic resistivity) depends on δ, which has not been interpreted before. We have shown that the densities of states are very similar for various $R_{5}$Rh${}_{6}$Sn${}_{18}$ compounds, and they practically do not depend on the metal *R*, while they are determined by the Rh *d*-and Sn *s*- and *p*-electron states. The band structure calculations for Sc${}_{5}$Rh${}_{6}$Sn${}_{18}$ have not been reported yet. We also found that the electronic specific heat coefficients $\gamma_{0}$ for the stoichiometric samples were always larger with respect to the $\gamma_{0}$ of the respective samples with vacancies at the *R* sites, which correlates with the results of ab initio calculations.

## 1. Introduction

The field cubic skutterudite-like compounds of the formula $R_{3}M_{4}$Sn${}_{13}$, where *M* is a *d*-electron-type metal and *R* is extended by a Ce or Yb, or their tetragonal equivalents $R_{5}M_{6}$Sn${}_{18}$, have been known as a family of strongly correlated electron systems (SCESs), attracting a great deal of attention for the past decade because of their unique low-temperature characteristics [1,2,3] resulting either from *f*- or *d*-electron correlations, as well as for their promising thermoelectric properties [4] due to the existence of Sn structural cages filled with Sn, *R*, and/or *M* atoms, respectively. In recent reports, we documented, however, that the strong covalent bonding characteristic of these materials excludes the rattling effect in these cages [5], which causes the observed value of figure of merit ZT to be much smaller than expected. Here, we focus on the electronic structure of the 5:6:18 La-, Lu-, and Sc-based superconductors.

Superconducting stannides of the type $R_{5}$Rh${}_{6}$Sn${}_{18}$ (*R* = Sc, Y, Lu) were first reported by Remeika et al. [6]. These compounds have a cage-like structure, crystallise in the tetragonal structure (space group I$4_{1}/acd$) [7], and exhibit conventional BCS-type superconductivity below 5 K (Sc [6,8,9]), 3 K (Y [6,10,11]), and 4 K (Lu [6,11,12]), respectively. Numerous measurements indicate for these quasiskutterudites that disorder with the presence of atomic scale disorder, structural defects, static atomic displacements, vacancies, or local inhomogeneity over the length scale of the coherence length ξ generate an enhancement of the superconducting transition temperature $T_{c}$. This enhancing of superconductivity by atomic disorder was investigated in detail in our series of previous reports [13,14,15,16,17,18].

In addition, the vacancies δ in atomic positions occupied by *R* determine quite different behaviours in the normal state resistivity ρ(T) of each $R_{5-\delta }$Rh${}_{6}$Sn${}_{18}$ sample. These various electrical transport properties were explained as the result of different stoichiometry of the respective samples. Namely, the resistivity measurements of all off-stoichiometric $R_{5-\delta }$Rh${}_{6}$Sn${}_{18}$ compounds (δ≠0) show semimetallic behaviour in the normal state before becoming superconducting below $T_{c}$, while for the equivalent but stoichiometric $R_{5}$Rh${}_{6}$Sn${}_{18}$ samples, the characteristics ρ(T) were reported to be typical of metals [15,16,17]. Energy-dispersive X-ray spectroscopy (EDXS) measurements confirm a deficiency of Lu or Y and excess Sn in the Lu${}_{5}$Rh${}_{6}$Sn${}_{18}$ and Y${}_{5}$Rh${}_{6}$Sn${}_{18}$ samples [12,16,17]. The stoichiometry from microanalysis is in agreement with the structural refinements [7,16,19,20], giving the composition (Sn${}_{1-x}R_{x}$)$R_{4}$Rh${}_{6}$Sn${}_{18}$, where the site with Wyckoff position 8b, which would be occupied by an *R* atom in the nominal $R_{5}$Rh${}_{6}$Sn${}_{18}$, is instead partially occupied by Sn atoms as a statistical mixture of Sn and *R*. We have experimentally demonstrated that the off-stoichiometric samples show a semimetallic nature; exhibit a negative coefficient in resistivity (TCR), dρ/dT<0, in a wide temperature range; and obey Mott’s law $\rho \propto \exp[{\left(\frac{\Delta_{M}}{k_{B}T}\right)}^{1/4}]$, while the respective more stoichiometric samples with δ≈0 are metallic ($\Delta_{M}$ is the width of the pseudogap). Similarly, the normal-state electronic specific heat coefficient $\gamma_{0}^{\left(n\right)}$ is always measured larger for the *metallic* samples in comparison to $\gamma_{0}^{\left(n\right)}$ of the more off-stoichiometric equivalent ones. Understanding of the various properties of $R_{5}$Rh${}_{6}$Sn${}_{18}$ depending on their stoichiometry requires calculations of their electronic structure. Here, we report the band structure calculations for the series of $R_{5}$Rh${}_{6}$Sn${}_{18}$ compounds and discuss their electrical transport properties. For Sc, the ab initio calculations are presented for the first time.

The work is a review, and its aim is to show how much the electronic structure of the system can be changed in the presence of vacancies δ, especially near the Fermi level, and how this change affects electric transport at $T>T_{c}$ and the Sommerfeld coefficients. The enhancement of $T_{c}$ is due to the presence of the lattice defects we studied previously [13,14,15,16,17,18]. The ab initio calculations for Y${}_{5-\delta }$Rh${}_{6}$Sn${}_{18}$ and Lu${}_{5-\delta }$Rh${}_{6}$Sn${}_{18}$ (δ=0,0.5) have been reported in [15,17] and are here compared with that performed for Sc sample. Within the series of 5:6:18 compounds, we obtained very similar bands below $\epsilon_{F}$ for the Sc, Y, and Lu compounds. The most important result is the pseudogap located at about −0.3 eV below the Fermi level, which easily can be moved towards $\epsilon_{F}$ by vacancies δ. Based on this observation, we interpret the semimetallicity of $R_{5-\delta }$Rh${}_{6}$Sn${}_{18}$ (*R* = Sc, Y, and Lu) in their normal states at $T>T_{c}$ and the δ-dependent Sommerfeld coefficients. A very similar dichotomy in the ρ(T) characteristics has already been observed in the filled skutterudite compounds (e.g., in CeRu${}_{4}$Sb${}_{12}$ [21]). This report could be useful in interpreting similar behaviours in several other intermetallics.

## 2. Experimental and Computational Details

The $R_{5}$Rh${}_{6}$Sn${}_{18}$ polycrystalline samples (*R* = Y and Lu) were obtained by arc melting technique and then annealed at 870 ${}^{\circ }$C for 2 weeks. The polycrystalline samples were checked by X-ray diffraction (XRD) analysis, using PANalytical Empyrean diffractometer equipped with Cu K$\alpha_{1,2}$ source, and found to have a tetragonal structure (space group I$4_{1}/acd$) [7,19]. Stoichiometry and homogeneity were checked by electron microprobe technique (scanning electron microscope JSM-5410 equipped with an energy-dispersive X-ray spectrometry microanalysis system).

Electrical resistivity ρ, magnetic susceptibility, and specific heat *C* were measured using a PPMS (physical properties measurement system) device.

The X-ray photoelectron spectroscopy (XPS) spectra were obtained with monochromatised Al $K_{\alpha }$ radiation at room temperature in a vacuum of $6\times 10^{-10}$ Torr using a PHI 5700 ESCA spectrometer. Each sample was broken under a high vacuum immediately before measuring the spectra.

The electronic band structures of the series of $R_{5}$Rh${}_{6}$Sn${}_{18}$ (*R* = Sc, Y, and Lu) compounds were calculated by the full-potential linearised augmented plane waves (FP-LAPW) method complemented with local orbitals (LO) [22] implemented in the WIEN2k computer code [23]. The scalar-relativistic Kohn–Sham approach was applied for the valence and semicore states using spin-orbit (SO) coupling accounted for by means of the second variational method [22], while the core states were treated within the fully relativistic density functional formalism. The generalised gradient approximation form (GGA) of the exchange-correlation energy function together with parametrisation (PBEsol) derived for solids by Perdew et al. [24] were applied. For the Lu 4f and Rh 4d band states of Lu${}_{5}$Rh${}_{6}$Sn${}_{18}$ and Lu${}_{4.5}$Rh${}_{6}$Sn${}_{18}$, the exchange correlation (XC) potential was corrected by the Hubbard-type correlation interaction using the LSDA+U method [25,26] with $U_{eff}$ equal to 6.8 eV for Lu 4f and 3.0 eV for Rh 4d states, respectively. For the remaining $R_{5}$Rh${}_{6}$Sn${}_{18}$ compounds (*R* = Y and Sc), where atoms *R* do not have 4f-electron states, the correlation energy $U_{d}=3$ eV was only included in the calculations of the Rh 4d states. The k-mesh was tested against the total energy convergence, and satisfactory precision of few meV was achieved with 7×7×7 mesh ($N_{k}=40\vec{k}$ vectors in irreducible Brillouin zone (BZ)). For all investigated compounds, the muffin-tin radius (R${}_{MT}$) of the same value (R${}_{MT}=1.27$ Å) was assumed for each atomic species. The ab initio calculations were carried out using the experimental lattice parameters (tetragonal structure, space group I$4_{1}/acd$) obtained from XRD for respective $R_{5}$Rh${}_{6}$Sn${}_{18}$ compounds. In each case, the atomic positions were relaxed using the Multisecant Rank One algorithm implemented in WIEN2k code.

The unit cell of the R${}_{5}$Rh${}_{6}$Sn${}_{18}$, with a local environment of R${}_{1}$ and R${}_{2}$ sites marked by polyhedrons, is shown in Figure 1. The primitive cell is built of four formula units and consists of 116 atoms located at 10 Wyckoff positions: Rh atoms split into 16(d) (Rh${}_{1}$) and 32(g) (Rh${}_{2}$) positions, and *R* into 8(b) ($R_{1}$) and 32(g) ($R_{2}$), while Sn atoms locate at 6 different Wyckoff positions.

It is worth noting that $R_{1}$ atoms, surrounded only by Sn ones (cf. Figure 1a), are isolated from other constituent atoms (Rh, *R*). The nearest neighbouring Sn atoms are located at a distance not less then 3.3 Å. On the contrary, the $R_{2}$ atoms are tightly bound to Rh${}_{1}$, Rh${}_{2}$, and Sn${}_{6}$ atoms, located at distances not greater than 3.01 Å, which form distorted tetrahedrons (Figure 1b). It is easily seen that these tetrahedrons are not isolated from each other but form an array connected alternatively by Rh and Sn${}_{6}$ atoms.

## 3. Electronic Structure of $R_{5}$Rh${}_{6}$Sn${}_{18}$ (R = Sc, Y, Lu): Experiment and Ab Initio Calculations

Figure 2 compares the valence band (VB) XPS spectra of Y${}_{5}$Rh${}_{6}$Sn${}_{18}$ and Lu${}_{5}$Rh${}_{6}$Sn${}_{18}$ with the respective total (T) density of states (DOS). The TDOS for Sc${}_{5}$Rh${}_{6}$Sn${}_{18}$ are also shown. The valence band XPS spectra shown in the figure are almost identical and are dominated by Rh *d*-electronic states. For the Lu-sample, the *s*, *p*, and *d* electronic valence band states are clearly overlapped by the SO 4f-electron states of Lu. Similarly, the calculated valence bands of $R_{5}$Rh${}_{6}$Sn${}_{18}$ have a very similar structure and correlate well with the VB XPS spectra, as shown in Figure 2. The Lu 4f-electron XPS states are well accounted for by the Lu 4f states calculated for $U_{f}=6.8$ eV, which signals that the density-functional theory (DFT) calculations with $U_{f}=6.8$ eV and accompanying $U_{d}=3$ eV are correct for obtaining the details in the electronic bands of Lu${}_{5}$Rh${}_{6}$Sn${}_{18}$ near the Fermi level ($\epsilon_{F}$) as well as the value of TDOS, $2N(\epsilon_{F})$, at $\epsilon_{F}$ ($N(\epsilon_{F})$ is the total DOS at the Fermi level for one spin direction and is $\frac{1}{2}$ of TDOS($\epsilon_{F})$ for the paramagnetic system). For the Y- and Sc-compounds, the Rh-*d* electron correlations with $U_{d}=3$ eV were always taken into account in the same way, neglecting the effect of occupation of the *f*-electron shell.

In order to interpret the contribution of various electronic states in TDOS and locate them properly with respect to the Fermi level, one should present the energy distribution of the partial states of each atom. Such a comparison of various atomic states was shown earlier for Y${}_{5}$Rh${}_{6}$Sn${}_{18}$ in [27] and for Lu${}_{5}$Rh${}_{6}$Sn${}_{18}$ in [16]. Figure 3 displays the summarised electronic *s*, *p*, and *d* VB states of Sc${}_{5}$Rh${}_{6}$Sn${}_{18}$ for Sc, Rh, and Sn atoms. When comparing the calculated band structures of the series of $R_{5}$Rh${}_{6}$Sn${}_{18}$ isostructural and isoelectronic compounds, one finds that: (i) their TDOS are very similar, and thus their VB XPS bands are also comparable; (ii) the VB XPS bands are dominated by $4d^{8}$ Rh states; (iii) a distinct pseudogap is located ∼0.3 eV below $\epsilon_{F}$; and finally (iv), the VB electronic states of atom *R* and Sn, as well as Sc and Rh, are strongly hybridised, as shown in Figure 4. The hybridisation is the strongest between the valence bands of Sn${}_{6}$ and $R_{1}$ and $R_{2}$ electronic states, which correlates well with the largest electronic charge transfer to interstitial space, ΔQ, in these atoms (cf. Table 1). We also note that the maximum value of ΔQ is accompanying the largest interatomic distances $d_{nn}$ between these atoms. This means that the $R_{1}$ atom is well bounded in the Sn cage, which eliminates the expected rattling effect for these materials. Very similar covalent bounding due to charge transfer between atom *R* and Sn was also calculated for the series of isoelectronic La${}_{3}M_{4}$Sn${}_{13}$ and Ce${}_{3}M_{4}$Sn${}_{13}$ (*M* = Co, Rh, and Ru) quasiskutterudites [28].

Regarding point (iv), strong hybridisation between the electronic states usually causes their delocalisation. In the case of the bonding of atoms $R_{1}$ (Sc, Y, Lu) and Sn${}_{6}$, one observes the largest distance $d_{nn}$ between them, and similarly, the accompanying charge transfer ΔQ is the largest (cf. Table 1). These observations, in contrast to the hybridisation effect, justify a greater localisation of electronic states for both the metal *R* (especially when *R* = Y and Lu) and Sn. Strong charge transfer ΔQ is typical of ionic and/or covalent bonds; moreover, the bond extension can also justify the mechanism of localisation due to Mott criteria [29]. For the series of 5:6:18-type compounds, it seems reasonable to explain that the *R* atom centred in the Sn cage forms strongly hybridised states with an environment, in competition with the localisation effect caused by the size of the cage and charge transfer effect. The formation of tetrahedral coordination of $R_{2}$ atoms (Figure 1b), typical for sp materials like Si or Ge, can be associated also with strong hybridisation of the valence states of Rh${}_{1}$, Rh${}_{2}$, Sc${}_{2}$, and Sn${}_{6}$, as shown in Figure 4a for an energy range between −1.2 eV and $\epsilon_{F}$. Relatively large electronic charge leakage from Sn${}_{6}$ atoms may also indicate enhanced hybridisation within the tetrahedron complex.

The VB XPS spectra shown in Figure 2 exhibit two peaks. The first one observed between $\epsilon_{F}$ and −5.5 eV is attributed to presence of the Rh 4d and also Sn 5p states, while for binding energies (−3 to −11 eV), the Sn 5s states provide a significant contribution to the intensity of the valence band spectra. One notes that the electronic structure calculated for the analogous cubic La${}_{3}$Rh${}_{4}$Sn${}_{13}$ and Ca${}_{3}$Rh${}_{4}$Sn${}_{13}$ [18] quasiskutterudites is very similar to that obtained for the 5:6:18 superconductors, except for the hybridisation pseudogap near $\epsilon_{F}$, which is not present in the case of the 3:4:13 superconducting materials. Similarly, the VB XPS spectra of the cubic 3:4:13 analogues are determined by the Rh 4d electronic states and Sn *p* and *s* valence electrons states.

The most important numerical results of the performed PBEsol + U calculations are listed in Table 2.

The Sommerfeld coefficient $\gamma_{0}^{calc}=\frac{\pi^{2}k_{B}^{2}N_{A}}{3}DOS\left(\epsilon_{F}\right)$ is compared with the $\gamma_{0}^{\left(SC\right)}$ measured under magnetic field $B>B_{c}$ at $T<T_{c}\left(0\right)$. When $\gamma_{0}^{\left(n\right)}$ is measured for $T>T_{c}$ at the zero magnetic field, its obtained value is much lower than the calculated one. Usually, similar $\gamma_{0}$ discrepancies are attributed to a vortex effect under external magnetic field. For a classic isotropic *s*-wave superconductor, the specific heat in the superconducting phase is dominated by that of the vortex cores, the number of which is proportional to *B*. In consequence, $\gamma_{0}^{\left(SC\right)}$ is field dependent when T→0. For Sc and Y samples, $\gamma_{0}^{\left(SC\right)}\propto B^{1/2}$ [33], indicating an energy gap without nodes; while for Lu sample, $\gamma_{0}^{\left(SC\right)}\propto B$ [10,11], which points to an isotropic superconducting gap.

The significant observations are: (i) the obtained coefficients $\gamma_{0}^{\left(n\right)}$ measured for samples with vacancies δ at zero magnetic field are much lower than expected; (ii) $\gamma_{0}^{\left(n\right)}$s for stoichometric samples are much larger than those obtained for the off-stoichiometric one. Both observations suggest a strongly decreased value of the DOS at $\epsilon_{F}$ in the case of δ=0.5. This behaviour would be possible after the pseudogap shift towards the Fermi level. We recently documented the predicted reconstruction of the DOS near the Fermi level for the off-stoichiometry system Lu${}_{5-\delta }$Rh${}_{6}$Sn${}_{18}$ with δ=0.5 [16]. The ab initio calculations documented the shift of pseudogap in DOS of Lu${}_{4.5}$Rh${}_{6}$Sn${}_{18}$ by 0.1 eV towards $\epsilon_{F}$ with respect to the stoichiometric sample, while the DOS located at lower binding energies are practically unchanged. In consequence of the calculated change in DOS at narrow energy range near $\epsilon_{F}$, the semimetallic nature of the electrical resistivity, experimentally obtained for the $R_{5-\delta }$Rh${}_{6}$Sn${}_{18}$ samples, can be interpreted as a result of the number of vacancies at *R* sites, while the stoichiometric equivalents are metallic (see Section 4).

It is also worth noting that taking into account the electron–phonon coupling parameter λ∼0.5 [31] and mass enhancement factor due to electron–electron interactions $\mu^{★}∼0.1$, the renormalised DOS, $2N\left(\epsilon_{F}\right)\times \left(1+\lambda +\mu^{★}\right)$, gives $\gamma_{0}^{\left(n\right)}$s for the stoichiometric samples *R* = Sc, Y, and Lu very close to the calculated $\gamma_{0}^{calc}$s, while the $\gamma_{0}^{\left(n\right)}$ for this procedure still remains much smaller than the calculated value for the samples with vacancies (cf. Table 1).

Our recent studies were focused on the skutterudite-related compounds with atomic scale disorder leading to the appearance of a novel high-temperature superconducting state with the critical temperature $T_{c}^{★}>T_{c}$ [13]. The $R_{5}$Rh${}_{6}$Sn${}_{18}$ compounds are the reference materials, which when doped, show an increase in $T_{c}$. Several experimental and theoretical attempts have been undertaken to answer the question of why $T_{c}$ is enhanced when the amount of disorder increases (e.g., [13,15,34]). Our experimental observations allowed us to propose a phenomenological model that explains an increase in $T_{c}$ by stronger stiffening of the locally inhomogeneous phase [15] due to doping. It seems interesting to show how much the doping of the parent $R_{5}$Rh${}_{6}$Sn${}_{18}$ samples (also having local atomic disorder over the length scales of the coherence length ξ) changes their electronic structure. We therefore calculated the TDOS for a number of various dopants; one result is presented in Figure 5 for Y${}_{5}$Rh${}_{6}$Sn${}_{18}$ for doping with Sr. It can be seen that the doping of the superconducting matrix does not practically change its TDOS, and the change of the DOS at the Fermi level is negligibly small. Hence, the presence of high-temperature inhomogeneous phase ($T_{c}^{★}$) with coexistance of the bulk superconducting one (bulk phase $T_{c}$) is most probably attributed to the local atomic disorder, which generates larger lattice stiffening. For the known quasiskutterudite compounds, the electron–phonon coupling parameter $\lambda^{★}$ obtained for the inhomogeneous $T_{c}^{★}$ superconducting phase is always larger than λ of the respective bulk $T_{c}$ superconducting state (cf. [31]). Simultaneously, we documented experimentally that the observed pressure coefficients $|\frac{dT_{c}^{★}}{dP}|$ are larger as those of $T_{c}$. The primary reason for $|\frac{dT_{c}^{★}}{dP}|>|\frac{dT_{c}}{dP}|$ is the pressure dependence of the Debye temperature $\theta_{D}$, which leads to enhanced lattice stiffening in the disordered $T_{c}^{★}$ phase. This observation also justifies the Grüneisen parameter $\Gamma^{★}$ of the locally inhomogeneous phase larger than Γ of the bulk $T_{c}$ phase, as was reported in [31]. The Γs can be estimated from the expression [35,36]
(1)$$\frac{d\ln[N\left(\epsilon_{F}\right)U]}{d\ln V}\equiv \left(2\Gamma_{G}-\frac{4}{3}\right)\frac{\lambda }{1+\lambda }\frac{1+\mu^{★}}{\lambda -\mu^{★}}$$
and McMillan relationship [37]
(2)$$T_{c}=\frac{\theta_{D}}{1.45}\exp\left\{\frac{-1.04(1+\lambda )}{\lambda -\mu^{*}\left(1+0.62\lambda \right)}\right\},$$
where $\mu^{*}$ is the Coulomb pseudopotential of Morel and Anderson [38] and electron–phonon coupling parameter [37,39]
(3)$$\lambda =\frac{N\left(\epsilon_{F}\right)\langle I^{2}\rangle }{M\langle \omega^{2}\rangle }.$$
To obtain the parameter Γ, the resistance measurements under external pressure and the ab initio calculation are useful (cf. [31]). One also notes that under external pressure, the change of $N(\epsilon_{F})$ is clearly documented in the calculations of the electronic bands.

## 4. Metallicity or Semimetallicity of $R_{5-\delta }$Rh${}_{6}$Sn${}_{18}$ (R = Sc, Y, Lu; δ≪1) and the Electronic Structure

As was mentioned above, the subtle reconstruction in the distribution of the TDOS around the Fermi level caused by vacancies could lead to drastic change in the normal-state electrical transport properties of the $R_{5-\delta }$Rh${}_{6}$Sn${}_{18}$ samples. The main reason for this band-structure effect is the deficiency of *R* indicated by energy-dispersive X-ray spectroscopy measurements, which is typical both for polycrystalline samples [17] and single crystals [12]. Our ab initio calculations documented that the vacancies at the *R*-sites move the pseudogap at 0.3 eV towards $\epsilon_{F}$. Then, one expects different characteristic of ρ(T) in the normal metallic state for the more stoichiometric sample with respect to the defected one. Figure 6 and Figure 7 show such a dichotomous behaviour for Y${}_{5-\delta }$Rh${}_{6}$Sn${}_{18}$ and Lu${}_{5-\delta }$Rh${}_{6}$Sn${}_{18}$, respectively.

The semimetallic nature of ρ(T) is known for single crystalline Sc${}_{5-\delta }$Rh${}_{6}$Sn${}_{18}$ from [12,33].

The $R_{5-\delta }$Rh${}_{6}$Sn${}_{18}$ samples with a semimetallic nature exhibit a negative temperature coefficient dρ/dT<0 in a large temperature range above $T_{c}$ (see Figure 6 and Figure 7). This anomalous increase in ρ with decreasing of *T* does not shows linear change, as could be expected for strongly disordered alloys [40], nor does it obey an activated law, while ρ(T) obeys Mott’s law $\rho \propto \exp[{\left(\frac{\Delta_{M}}{k_{B}T}\right)}^{1/4}]$, known as Mott variable-range hopping effect [41,42]. Here, $\Delta_{M}$ characterises the pseudogap in the band structure near the Fermi level. An agreement with Mott variable-range hopping behaviour was previously reported, for example, for some *d*-electron semiconducting Heusler alloys (c.f. ZrNiSn and TiNiSn [43,44,45]) and strongly correlated *f*-electron system Ce${}_{5}$RuGe${}_{2}$ [46].

## 5. Band Structure of the System of R${}_{5}$Rh${}_{6}$Sn${}_{18}$ Compounds, Comparison

Band structures shown in Figure 8 along high-symmetry *k* lines possess dispersive electronic states near the Fermi level, which is consistent with the metallic conductivity of $R_{5}$Rh${}_{6}$Sn${}_{18}$. The formation of the pseudogap below $\epsilon_{F}$ is a consequence of strong hybridisation between the band states located on various surfaces of the Brillouin zone, as shown in Figure 8 (*k*-lines: Y−Σ−Γ, Z−Σ1−N−P−Y1−Z). This valence band state can be moved to the Fermi level when the bands are calculated for the system with deficiency of *R*; this was documented recently for Lu${}_{4.5}$Rh${}_{6}$Sn${}_{18}$ [16].

The investigated compounds belong to a family of non-symmorphic materials. Previous investigations have shown that the bulk materials with non-symmorphic space groups may exhibit unusual properties of their electronic structure [47,48] coming from the band degeneracies entailed by non-symmorphic symmetry. A characteristic property of the electronic structure of such materials can be the presence of Dirac cones and Dirac nodes [47]. The band structures presented in Figure 8 reveal the appearance of several Dirac-cone-like shapes (e.g., on Γ−Z−Σ−N symmetry lines).

Figure 9 displays the band structure calculated along high-symmetry *k* lines in the Brillouin zone for stoichiometric Y${}_{4.5}$Sr${}_{0.5}$Rh${}_{6}$Sn${}_{18}$ sample. The aim of ab initio calculations was to show that the dopant (Sr) does not modify the band structure in the binding energies between $\epsilon_{F}$ and −0.5 eV, leaving the pseudogap at −0.3 eV. In effect, the resistivity of the Sr-doped sample is expected to be metallic; this is, however, not the case. The EDXS measurements indicated the deficiency of Y in the Y${}_{4.5}$Sr${}_{0.5}$Rh${}_{6}$Sn${}_{18}$ sample too, which is a reason for the semimetallic behaviour in ρ(T) data and the Mott’s $ln\rho ∼T^{-1/4}$ dependence [17].

## 6. Conclusions

The dominant role of the vacancies and structural defects in the series of $R_{5-\delta }$Rh${}_{6}$Sn${}_{18}$ BCS superconductors can be observed in the increasing in $T_{c}$, as well as in different electrical transport properties, which is metallic (δ≈0) or semimetallic (δ∼0.5) in nature in the normal state of the respective sample. This dichotomous behaviour is the result of various numbers of vacancies δ at the sites occupied by metal *R*. The metallic character of the 5:6:18-type quasiskutterudites is observed for the samples with a small number of vacancies. When the number of vacancies is increased, the sample is semimetallic with the Mott variable-range hopping effect. We present the band structure calculations for stoichiometric $R_{5}$Rh${}_{6}$Sn${}_{18}$ materials (*R* = Sc, Y). For Lu samples, the band structure calculations are performed either for the stoichiometric or off-stoichiometric samples. The ab initio calculations give very similar band structures for the series $R_{5}$Rh${}_{6}$Sn${}_{18}$ and indicate the narrow and deep pseudogap at about −0.3 eV, which can be moved toward the Fermi level in the defected sample. The presence of this hybridisation pseudogap in the bands near $\epsilon_{F}$ allows one to interpret the various physical properties observed in similar $R_{5-\delta }$Rh${}_{6}$Sn${}_{18}$ samples, determined by the vacancies δ. The off-stoichiometric samples exhibit a negative coefficient in resistivity, $\frac{d\rho }{dT}<0$, in a wide temperature range of about 200 K, and obey Mott’s law $\rho \propto \exp[{\left(\frac{\Delta_{M}}{k_{B}T}\right)}^{1/4}]$, while the respective more stoichiometric counterparts are metallic. Both behaviours are well documented by the DFT calculations. In addition, the observation of Sommerfeld coefficients $\gamma_{0}^{\left(n\right)}$ smaller than the calculated one is characteristic of the $R_{5-\delta }$Rh${}_{6}$Sn${}_{18}$ series, and results from reconstruction of the bands near the Fermi level due to the presence of vacancies δ.

## Figures and Tables

**Figure 1 materials-15-02451-f001:**
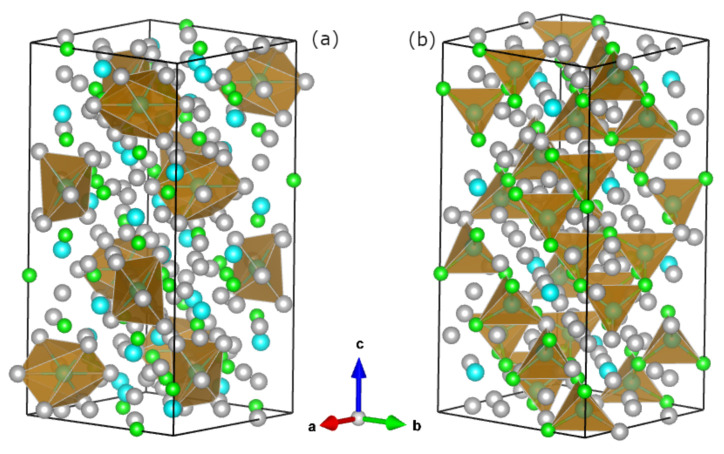
The unit cell of $R_{5}$Rh${}_{6}$Sn${}_{18}$ compound. The polyhedra show the nearest local environment of the $R_{1}$ (**a**) and $R_{2}$ (**b**) sites. The grey, green, and blue spheres represent Sn, Rh, and *R* atoms (*R* = Sc, Lu, Y), respectively.

**Figure 2 materials-15-02451-f002:**
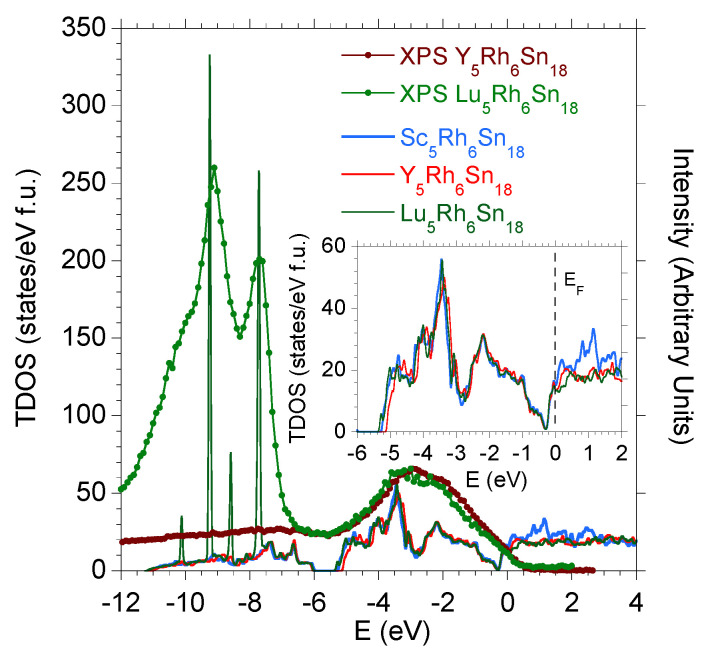
Valence band XPS spectra for Y${}_{5}$Rh${}_{6}$Sn${}_{18}$ (brown points) and Lu${}_{5}$Rh${}_{6}$Sn${}_{18}$ (green points) are compared with the TDOS calculated for $R_{5}$Rh${}_{6}$Sn${}_{18}$ (*R* = Sc, Y, Lu) within the LSDA approximation. The intensities of the measured VB XPS spectra are renormalised to the same background either of the *R* = Y or *R* = Lu samples at energies $E>\epsilon_{F}$.

**Figure 3 materials-15-02451-f003:**
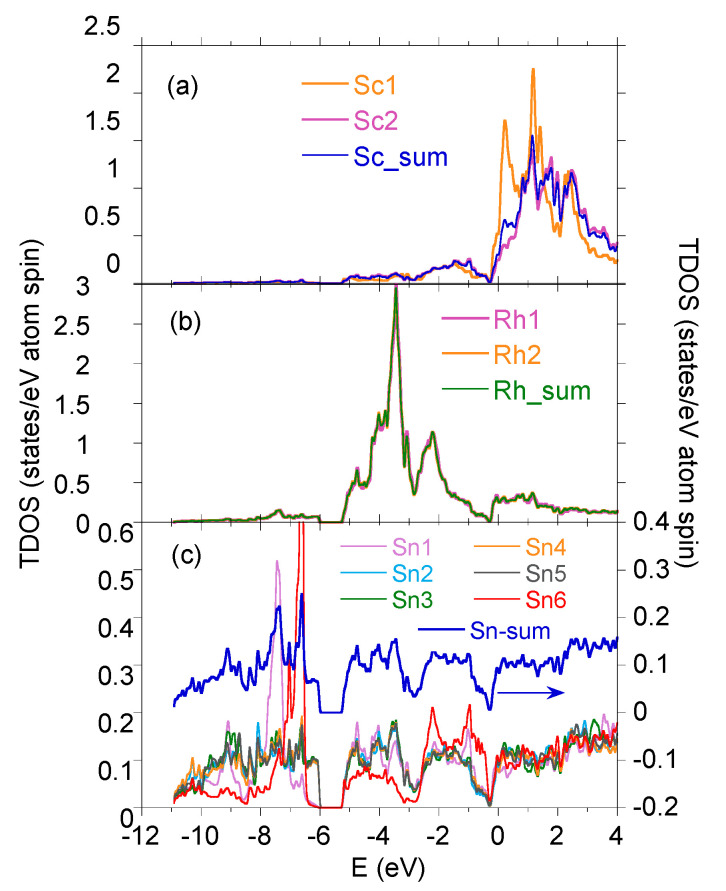
Total DOS per atom (Sc (**a**), Rh (**b**), Sn (**c**)) in Sc${}_{5}$Rh${}_{6}$Sn${}_{18}$.

**Figure 4 materials-15-02451-f004:**
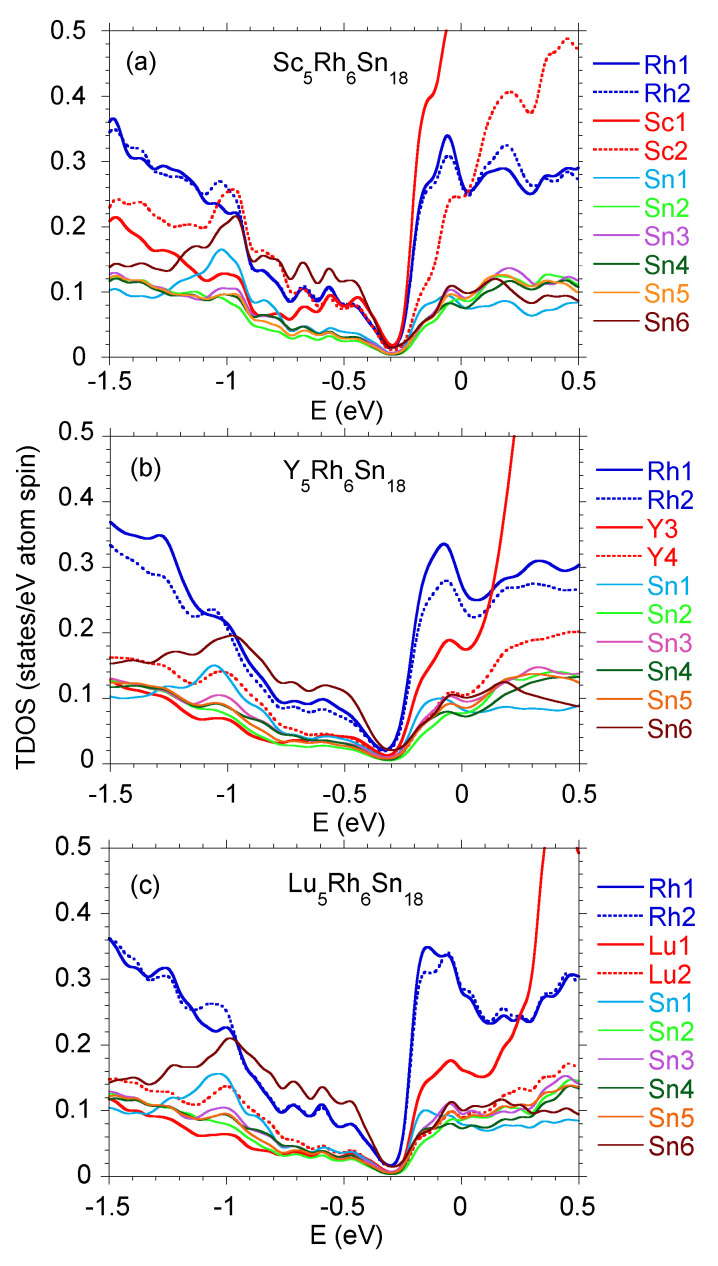
Partial (total) DOS per one atom for Sc${}_{5}$Rh${}_{6}$Sn${}_{18}$ (**a**), Y${}_{5}$Rh${}_{6}$Sn${}_{18}$ (**b**), and Lu${}_{5}$Rh${}_{6}$Sn${}_{18}$ (**c**), comparison.

**Figure 5 materials-15-02451-f005:**
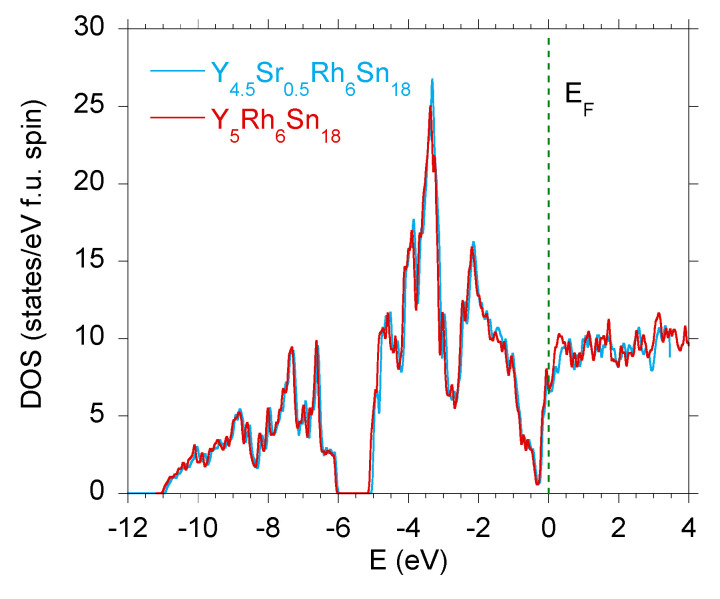
Total DOS calculated for paramagnetic Y${}_{5}$Rh${}_{6}$Sn${}_{18}$ compared with the TDOS for Y${}_{4.5}$Sr${}_{0.5}$Rh${}_{6}$Sn${}_{18}$.

**Figure 6 materials-15-02451-f006:**
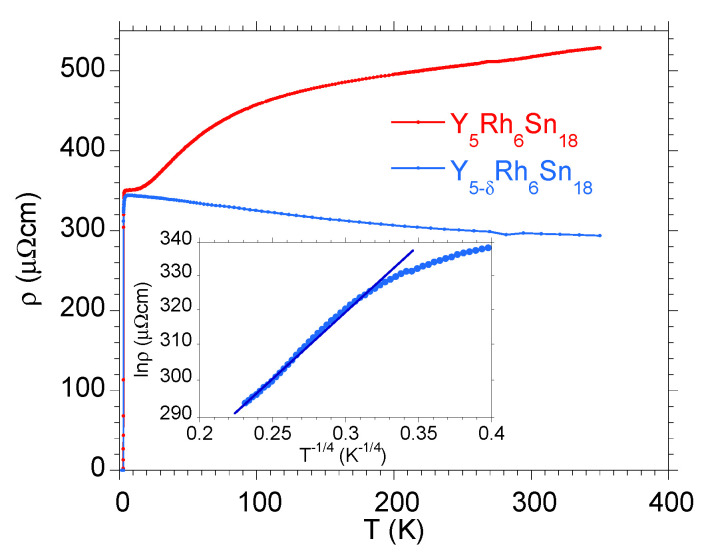
Electrical resistivity ρ(T) for stoichiometric Y${}_{5}$Rh${}_{6}$Sn${}_{18}$ (red points) and off-stoichiometric Y${}_{5-\delta }$Rh${}_{6}$Sn${}_{18}$ (blue points) samples (δ=0.4±0.11). Inset displays the resistivity data for Y${}_{5-\delta }$Rh${}_{6}$Sn${}_{18}$ in coordinates $\ln\rho =f(T^{-1/4})$. The linear lnρ vs. $T^{-1/4}$ behaviour is observed between ∼100 K and 350 K.

**Figure 7 materials-15-02451-f007:**
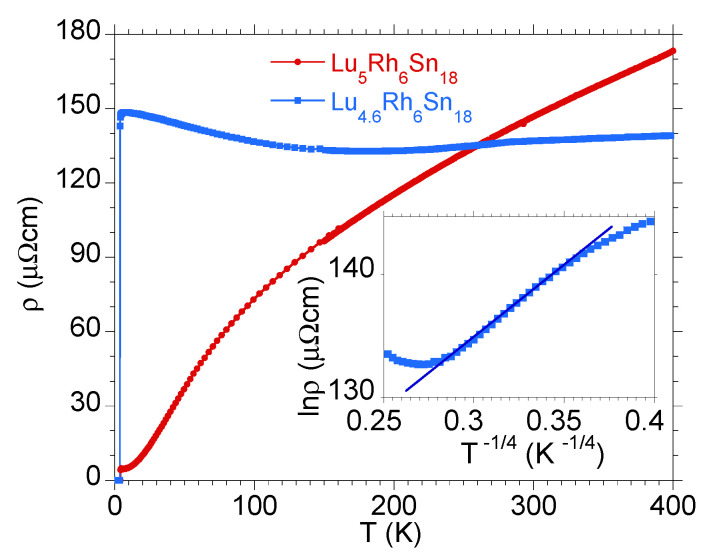
Electrical resistivity ρ(T) for Lu${}_{5}$Rh${}_{6}$Sn${}_{18}$ (red points) and Lu${}_{4.6}$Rh${}_{6}$Sn${}_{18}$ (blue points). Inset displays the resistivity data for Lu${}_{4.6}$Rh${}_{6}$Sn${}_{18}$ in coordinates $\ln\rho =f(T^{-1/4})$ and the linear lnρ vs. $T^{-1/4}$ behaviour between ∼60 K and ∼160 K.

**Figure 8 materials-15-02451-f008:**
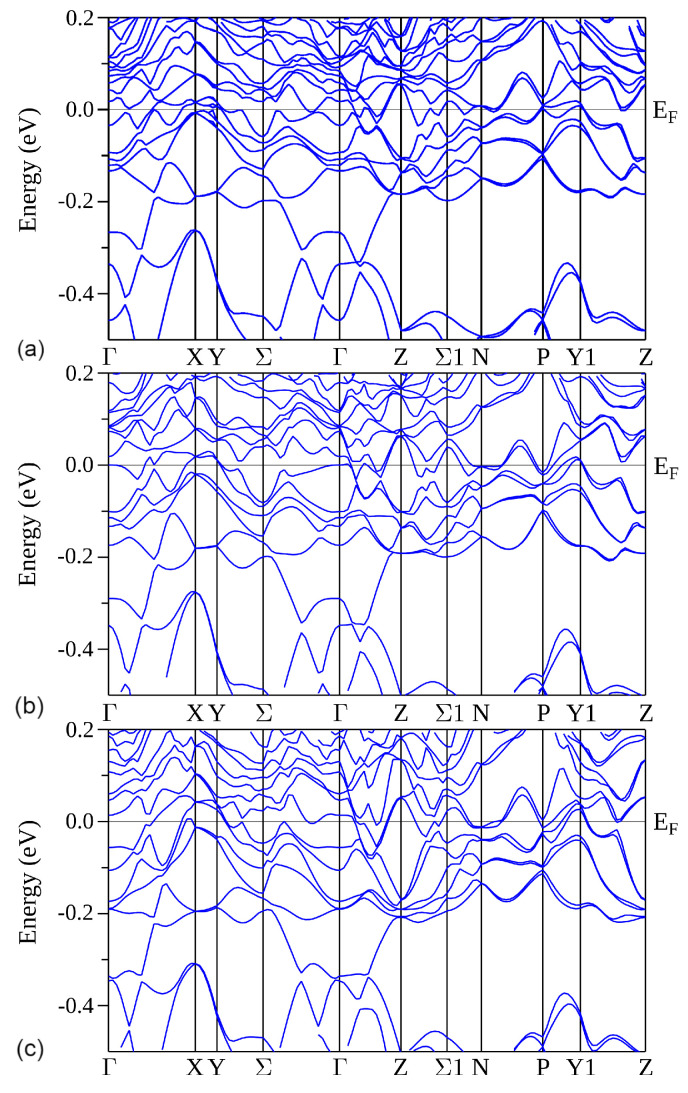
The band structure calculated along high symmetry *k* lines in the Brillouin zone of Sc${}_{5}$Rh${}_{6}$Sn${}_{18}$ (**a**), Y${}_{5}$Rh${}_{6}$Sn${}_{18}$ (**b**), and Lu${}_{5}$Rh${}_{6}$Sn${}_{18}$ (**c**). The calculations were performed for $U_{d}=3$ eV (for Lu${}_{5}$Rh${}_{6}$Sn${}_{18}$, $U_{d}=3$ eV) and $U_{f}=6.8$ eV.

**Figure 9 materials-15-02451-f009:**
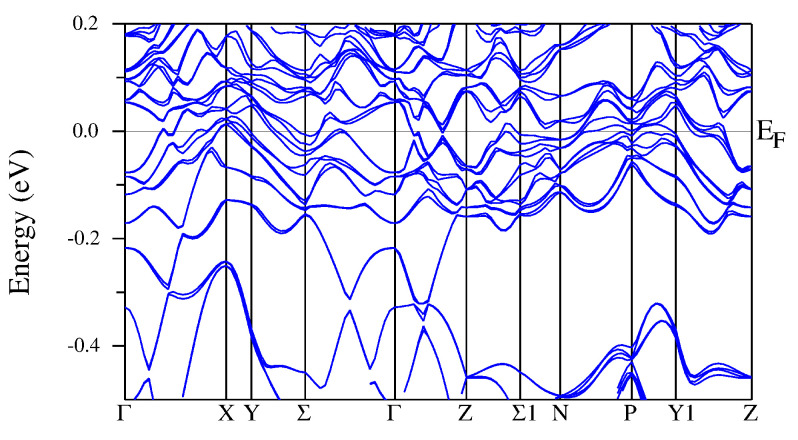
The band structure calculated along high-symmetry *k* lines in the Brillouin zone of Y${}_{4.5}$Sr${}_{0.5}$Rh${}_{6}$Sn${}_{18}$. The calculations were performed for $U_{d}=3$ eV.

**Table 1 materials-15-02451-t001:** Atomic local properties—distance to nearest neighbour $d_{nn}$ and charge leakage from atomic muffin-tin spheres ΔQ. Position No. enumerates Wyckoff positions, the coordinates of which are given in Table 1) of [27]. Subscripts in atomic labels indicate different Wyckoff positions.

Atom			Sc${}_{5}$Rh${}_{6}$Sn${}_{18}$		Y${}_{5}$Rh${}_{6}$Sn${}_{18}$		Lu${}_{5}$Rh${}_{6}$Sn${}_{18}$	
Type	Position No. Ref. [27]	Label	$d_{\mathrm{nn}}$ (Å)	ΔQ (e)	$d_{\mathrm{nn}}$ (Å)	ΔQ (e)	$d_{\mathrm{nn}}$ (Å)	ΔQ (e)
Rh	1	Rh${}_{1}$	2.62 (Sn)	1.283	2.65	1.319	2.64	1.300
Rh	2	Rh${}_{2}$	2.62 (Sn)	1.282	2.63	1.316	2.63	1.300
Sc/Y/Lu	3	$R_{2}$	3.33 (Sn)	2.009	3.38	2.630	3.36	2.617
Sc/Y/Lu	4	$R_{1}$	2.95 (Rh)	1.910	3.00	2.520	2.99	2.497
Sn	5	Sn${}_{1}$	2.69 (Rh)	2.306	2.74	2.323	2.72	2.305
Sn	6	Sn${}_{2}$	2.62 (Rh)	2.306	2.64	2.314	2.64	2.308
Sn	7	Sn${}_{3}$	2.62 (Rh)	2.304	2.66	2.311	2.64	2.305
Sn	8	Sn${}_{4}$	2.62 (Rh)	2.295	2.65	2.302	2.64	2.296
Sn	9	Sn${}_{5}$	2.62 (Rh)	2.305	2.63	2.312	2.63	2.306
Sn	10	Sn${}_{6}$	3.01 (R)	2.359	3.065	2.348	3.05	2.344

**Table 2 materials-15-02451-t002:** Structural characterisation (lattice parameters *a* and *c*), total DOS at the Fermi level, and the electronic specific heat coefficient $\gamma_{0}^{\left(SC\right)}$ obtained under magnetic field of 4 T at $T<T_{c}$ and $\gamma_{0}^{\left(n\right)}$ in normal state ($T>T_{c}$, B=0), respectively, from the C/T vs. $T^{2}$ linear dependence at T=0. The Sommerfeld coefficients in brackets are assigned to the off-stoichiometry samples with vacancies δ, respectively.

	Y${}_{5}$Rh${}_{6}$Sn${}_{18}$	Sc${}_{5}$Rh${}_{6}$Sn${}_{18}$	Lu${}_{5}$Rh${}_{6}$Sn${}_{18}$
*a* (Å)	13.7601 [17]	13.5826 [30]	13.7590 [16]
*c* (Å)	27.5412 [17]	27.1504 [30]	27.4747 [16]
DOS at $\epsilon_{F}$ (1/eV f.u.)	13.20	14.81	11.50
$\gamma_{0}^{calc}$ (mJ/mol K${}^{2}$) (B=0)	31.2	34.90	27.20
$\gamma_{0}^{\left(SC\right)}$ (mJ/mol K${}^{2}$) (B=4 T)	39 (24) [27]	(35.2) [30]	36 (47) [16]
$\gamma_{0}^{\left(n\right)}$ (mJ/mol K${}^{2}$) (B=0)	12.1 (9) [31]	(∼7) [32]	18 (5.5) [31]
$\gamma_{0}^{calc}\times \left(1+\lambda +\mu^{★}\right)$ (mJ/mol K${}^{2}$)	19.4 (14.4)	(11.2)	28.8 (8.8)

## Data Availability

Data supporting reported results are not available.

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
