# Peer review of "Band Structure Studies of the *R*_5_Rh_6_Sn_18_ (*R* = Sc, Y, Lu) Quasiskutteridite Superconductors"

_materials, 2022, doi:10.3390/ma15072451_

Round 1

Reviewer 1 Report

Review of; "Electronic structure studies of the R5Rh6Sn18 (R = Sc, Y, Lu) quasiskutteridite superconductors",
by J Deniszczyk and A Ślebarski.

The manuscript presents an overview of experimental and calculated electronic properties of selected members of the R5Rh6Sn18 series of compounds, with new results on band structure calculations for the Sc-based compound.

I found this an interesting manuscript to read. The juxtaposing of the electrical resistivity against band structure results is thought provoking, and I expect this manuscript will invite further related studies. For this class of materials, an insightful and somewhat unexpected finding is reached from the authors' XPS results concerning the proposed absence of rattling motion in the R1 atom. The striking results of a fundamental change in the nature of the electrical resistivity in the normal state induced by small changes in defect-related disorder of the R atom is extraordinary and of much interest. With a view to this point, I was expecting to see a graph displaying the resistivity against temperature for Sc5Rh6Sn18 in addition to those of the Lu- and Y-based compounds in figure 8, and hence I recommend the authors to revisit the compilation of figure 8 in this regard.

The text is easy to follow and the manuscript is logically organized. The presentation makes a compelling case that I believe would be of much interest to the community, and therefore I recommend the paper to be considered for publication.

Before finalizing the manuscript, the authors are invited to respond to the following points, listed in order of appearance.        

p2, line 39: correct the spelling of "metallic".
p2, line 49: correct "Cu Kα1,2 a source," - delete "a".
p2, line 50: for consistency, change the square bracketing to round brackets.
p2, line 69: change to; "...equal to...".
p2, line 75: change "specious" to "species".
p2, line 83: change "It worth note..." to "It is worth noting...".
p2, line 87: change "Easily to see..." to "It is easily seen...".
p3, line 92: define "DOS" where it is first used. This paragraph contains several more abbreviation symbols that need to be defined. Referring to "TDOS" in this paragraph, does this have a meaning different from "Total DOS" used in the caption of figure 5? This caption additionally also uses the notation "TDOS".  
p3, caption figure 2: "VB XPS spectra are renormalized to the background of one of the samples at energies E > ɛF.". Was it either of the R=Y or R=Lu compounds that was used for the background normalization? I think this should be stated in the caption.
p5, line 121: "In the case of the bonding of atom R1 (Sc, Y, Lu) and Sn one,..." - which of the six Sn sites are being referred to in this context?
p5, line 128: change to read; "...the R atom centered in the Sn cage...".
p5, line 129: "...in competition to location effect..." - should this be "localization" instead?
p5, line 132: correct the spelling of "between".
p5, line 140: in the comparison between the title compounds and the 3:4:13 quasiskutterudites, it is mentioned that; "...however, the hybridisation pseudogap near εF is not present.". The reader assumes that this absence refers to an attribute of the 3:4:13 compounds. It would be useful to clarify this in the sentence.
p5, line 144: What is the difference between the notation on page3, line 100; "N(ɛF)," and that on p5, line 144, "DOS(ɛF)"? Further, in Table 2 and elsewhere in the text a third variation of this notation; "DOS at ɛF" is used.
p6, line 144: "The Sommerfeld coefficient γ0 =...is well compared with the γ(SC)0 measured under magnetic field B > Bc at T < Tc(0). When γ0 is measured...". With reference to Table 2, none of the three types of gamma values listed in the Table are named "γ0" (they all have an additional superscript to differentiate between them). Likewise, the use of "γ0" in the text should be qualified for clarity. See also p9, line 218.
p6, line 155: suggest to change to; "...by 0.1 eV towards ɛF...".
p6, fourth line below figure 5 caption: suggest to change to read; "...which when doped...".
p7, first line: change to read; "...when it is doped...".
p7, line 6: correct the spelling of "quasiskutterudite".
p7, line 9: correct the spelling of "Simultaneously".
p7, line 161: change to; "...parameter Γ..."
p7, title section 4; correct the spelling of "semimetallicity".
p7, line 167: change to; "...drastic change in the...".
p7, line 172: "Then, one expects different characteristics ρ(T) in the normal metallic state for the more stoichiometric or defected sample." The meaning of this sentence is ambiguous. "more stoichiometric sample" seems to be the opposite of "defected sample" but the two are grouped together in this sentence as samples that show different ρ(T) characteristics in the normal-metal state.
p7, line 174: suggest to change to; "The semimetallic nature of ρ(T) is known...".
p7, line 176: change to; "...samples having semimetallic nature exhibit...".
p7, line 178: there are two instances here using a double negative which I believe ends up conveying the opposite to what the authors had  intended: Suggest to change to; "...T exhibits neither linear change, as could be expected for strongly disordered alloys [34], nor does it obey...".
p7, line 182: change to read; "...behavior was previously observed...".
p8, figure 6: Other than what is the case in figure 7, the caption and legend of figure 6 does not disclose what value of δ was used in Y5-δRh6Sn19.
p8, caption of figure 7: "Electrical resistivity ρ(T) for R5Rh6Sn18 [curve (a)] and Lu4.6Rh6Sn18 [curve (b)].". There are no labels (a) and (b) provided in the figure. Instead, resistivity data of stoichiometric and off-stoichiometric Lu5Rh6Sn18 are displayed.  
p8, line 187: change to read; "...conductivity of R5Rh6Sn18.".
p8, line 189: change to read; "...surfaces of the Brillouin zone,...".
p8, line 201: change to read; "...does not modify...".
p8, line 202: "...binding energies between ɛF and −0.5 eV, leaving the pseudogap at ∼ 0.3 eV.". Did the authors rather mean; "∼ —0.3eV"? Compare for example p9, line 217.
p8, line 203: change to read; "...expected to be metallic but this is,...".

end of report
___

Author Response

We are very thankful  for valuable critical remarks, both to Editor and  Referees. We took into account all critical comments. The manuscript  is improved, and it looks better. Below, we respond to critical comments of the  Referees.

Referee 1

We appreciate your critical comments, especially your suggestions on grammar and syntax.   All shortcomings and mistakes  are corrected. We also responded to important remarks:   

- p3, line 92: define "DOS" where it is first used. This paragraph contains several more abbreviation symbols that need to be defined. Referring to "TDOS" in this paragraph, does this have a meaning different from "Total DOS" used in the caption of figure 5? This caption additionally also uses the notation "TDOS".  
Authors: All abbreviation symbols are defined now (DOS, TDOS, XPS, VB, DFT…).

- p3, caption figure 2: "VB XPS spectra are renormalized to the background of one of the samples at energies E > ɛF.". Was it either of the R=Y or R=Lu compounds that was used for the background normalization? I think this should be stated in the caption.                                                                                                  Authors: In the caption to Fig. 2: The intensities of the measured VB XPS spectra are renormalized to the same background either of the  R=Y or  R=Lu samples at energies E > ɛF.

- p5, line 121: "In the case of the bonding of atom R1 (Sc, Y, Lu) and Sn one,..." - which of the six Sn sites are being referred to in this context?                                Authors: This is the R1 – Sn6 bonding.

- p5,line 140: in the comparison between the title compounds and the 3:4:13 quasiskutterudites, it is mentioned that; "...however, the hybridisation pseudogap near εF is not present.". The reader assumes that this absence refers to an attribute of the 3:4:13 compounds. It would be useful to clarify this in the sentence.                                                                                                                   Authors: We present now more clear sentence. “One notes, that the electronic structure calculated for the analogous  cubic La3Rh4Sn13 and Ca3Rh4Sn13 [13] quasiskutterudites is very similar to that obtained
for the 5 : 6 : 18 superconductors, except the hybridisation pseudogap near ɛF, which is not present in case of the 3 : 4 : 13 superconducting materials.”

- p5, line 144: What is the difference between the notation on page3, line 100; "N(ɛF)," and that on p5, line 144, "DOS(ɛF)"? Further, in Table 2 and elsewhere in the text a third variation of this notation; "DOS at ɛF" is used.                                                                                                                          Authors: Total DOS reflects the density of states for the both spin directions, while within the BCS theory more quantities are calculated for the DOS per one spin direction. For the paramagnetic superconducting materialse DOS for the up and down direction is the same.  N(ɛF) represents the DOS per one spin direction. (We comment it in Section 3).

- p6, line 144: "The Sommerfeld coefficient γ0 =...is well compared with the γ(SC)0 measured under magnetic field B > Bc at T < Tc(0). When γ0 is measured...". With reference to Table 2, none of the three types of gamma values listed in the Table are named "γ0" (they all have an additional superscript to differentiate between them). Likewise, the use of "γ0" in the text should be qualified for clarity. See also p9, line 218.                                                                                                                                Authors: According to the Referee's comment, the go coefficients from DFT calculations is named “calc” or from experimental data “exp”.

- p7, line 172: "Then, one expects different characteristics ρ(T) in the normal metallic state for the more stoichiometric or defected sample." The meaning of this sentence is ambiguous. "more stoichiometric sample" seems to be the opposite of "defected sample" but the two are grouped together in this sentence as samples that show different ρ(T) characteristics in the normal-metal state.
Authors: More clear sentence: “ Then, one expects different characteristic of ρ(T) in the normal metallic state for the more stoichiometric sample in respect to the defected one.”

- p8, figure 6: Other than what is the case in figure 7, the caption and legend of figure 6 does not disclose what value of δ was used in Y5-δRh6Sn19.                       Authors: d = 0.4±0.11, Fig. 6., figure caption.

- p8, caption of figure 7: "Electrical resistivity ρ(T) for R5Rh6Sn18 [curve (a)] and Lu4.6Rh6Sn18 [curve (b)].". There are no labels (a) and (b) provided in the figure. Instead, resistivity data of stoichiometric and off-stoichiometric Lu5Rh6Sn18 are displayed.                                                                                                     Authors: (Fig. 7), the figure caption is corrected.

With regards,

Andrzej Ślebarski

Reviewer 2 Report

In this study the authors reported on x-ray photoelectron spectroscopy and ab initio electronic structure investigations of the skutterudite-related R5Rh6Sn18 superconductors, where R= Sc, Y, and Lu. These compounds crystallise with a tetragonal structure (space group I41/acd) and are characterised by a deficiency of R atoms in formula unit (R5−δRh6Sn18, δ≪1). The vacancies at the R-sites as well as atomic local defects (often induced by doping) are a reason of enhancement in the superconducting transition temperature Tc of these materials, as well as metallic or semimetallic behaviours in their normal state, depending on the number of vacancies δ. i recommend the authors to improve the article by addressing the following issues

  1. clearly mention the objective and novelty of your work in abstract section;
  2. author should perform the proper spell check. Many punctuation errors are there in the file;
  3. The abstract should contain answers to the following questions: What problem was ‎studied and why is it important? What methods were used? What are the important results? What ‎conclusions can be drawn from the results? What is the novelty of the work and where does it go beyond ‎previous efforts in the literature? Please include specific and quantitative results in your abstract, while ‎ensuring that it is suitable for a broad audience;
  4. The physical significance of the conducted study should be provided for the understanding of researchers and readers;
  5. For validation of results, must compare present results for limited cases with already published articles;
  6. Several figures have been prepared and no physical presentation is provided;
  7. Why table 1 is prepared? What it shows? Clearly check table 2;
  8. Extend the conclusion section;
  9. Authors should mention the future recommendation in the last section;
  10. You should check your findings against previously published publications.

Author Response

We are very thankful  for valuable critical remarks, both to Editor and  Referees. We took into account all critical comments. The manuscript  is improved, and it looks better. Below, we respond to critical comments of the  Referees.

Referee2                                                                                                                    Thank you for your suggestions and critical comments, we hope that we have responded to them.

- Author should perform the proper spell check. Many punctuation errors are there in the file;

Authors: The text is corrected in terms of both grammar and syntax

-The abstract should contain answers to the following questions: What problem was ‎studied and why is it important? What methods were used? What are the important results? What ‎conclusions can be drawn from the results? What is the novelty of the work and where does it go beyond ‎previous efforts in the literature? Please include specific and quantitative results in your abstract, while ‎ensuring that it is suitable for a broad audience;

Authors: The Abstract is rewritten, we took into account the Referee's suggestions.

- The physical significance of the conducted study should be provided for the understanding of researchers and readers;                                                              - For validation of results, must compare present results for limited cases with already published articles;

Authors:  We agree with the Referee, the explanation is attached at the end of  Section:  Introduction.

The work is a review, its aim is to show how much the electronic structure of the  system can be changed in the presence of vacancies d, especially near the Fermi level, and how this change affects electric transport at T> Tc and the Smmmerfeld coefficients. The enhancement of Tc due to the presence of the lattice defects we  studied previously [References]. The ab initio  calculations for Y5-δRh6Sn18 and Lu5-δRh6Sn18 (δ =0, 0.5) have been reported in Refs.  [References] and are compared with that performed for Sc sample. Within the series of 5:6:18 compounds we obtained very similar bands  below єF,  the most important result is the  pseudogap located at about -0.3 eV below the Fermi level, which easily can be moved towards єF  by vacancies δ. Basing on this observation we interpret the semimetallicity of R5-δRh6Sn18 (R=Sc, Y, and Lu) in their “normal” states at T>Tc and the d-dependent Sommerfeld coefficients.      A very similar dichotomy in the ρ(T) characteristics has already been observed in the filled skutterudite compounds, e.g., in CeRu4Sb12 [Reference]. All the more, this report  can be useful in interpreting similar behaviors in several other intermetallics.”

- Several figures have been prepared and no physical presentation is provided;

Authors: It seems to us that all the figures  are described in the text together with the proper explanations.

- Why table 1 is prepared? What it shows? Clearly check table 2;

Authors: The intention of the authors is to show the correlation between the charge transfer and  the interatomic distances of respective atoms. In effect we expect strong covalent bonding between R atoms and Sn one, this bonding does not allowed the rattling effect. The Table 1 is supplemented now by Wyckoff positions. We also modified Table 2, as well as the description (text).

Below Table 2 we added comment: “The significant observations are: (i)  the coefficients γ0(n) measured for  samples with vacancies d at zero magnetic field are obtained much lower than they are expected, (ii)  γ0(n)s for stoichometric samples are much larger than these quantities obtained for  the off-stoichiometric one, the both observations  suggest  strongly decreased  value of the  DOS at єF in the case of δ= 0.5.                                                                   This behavior would be possible after the pseudogap shift forward to the Fermi level. We recently documented the predicted reconstruction of the DOS near the Fermi level for the off-stoichiometry system Lu5-δRh6Sn18  with δ<1 [Reference].  The ab initio calculations documented the shift of pseudogap in DOS of Lu5-δRh6Sn18 by 0.1 eV towards єF in respect to the stoichiometric sample, while the DOS located at lower binding energies are practically not changed.                      In consequence of the calculated change in  DOS at narrow energy range near єF, the  semimetallic  nature of the electrical resistivity,  experimentally documented for the  R5-δRh6Sn18  samples  can be interpreted as a result of  the number of vacancies at  R sites, while the stoichiometric equivalents are metallic (see Sec. 4).                                                                                                             It is also worth noting that taking  into account the electron-phonon coupling parameter λ~ 0.5  [Reference] and mass enhancement factor due to electron-electron interactions μ*~ 0.1,  the renormalized DOS,  2N(єF)x (1+ λ+μ*) gives  γ0(n) s for the stoichiometric samples R=Sc, Y, and Lu  very closed to the calculated γ0(calc)s, while this procedure still remains γ0(n)  much smaller than the calculated value for the samples with vacancies.”

- Extend the conclusion section;

Authors: Yes, the conclusion section is completed

Reviewer 3 Report

The paper is devoted to a quite interesting problem connected with electronic structure of R5Rh6Sn18 quasiskutteridite superconductors. Such objects are studied for several decades, but there are different problems which are still have to be solved, especially connected with the influence of the magnetic field. Research is based mostly on x-ray photoelectron spectroscopy methods, and some theoretical estimates are given.

The authors emphasize that the vacancies in atomic positions at the R-sites change the behaviour of the material, especially the resistivity. Authors present the results of measured electronic specific heat coefficient for zero and non-zero magnetic field and compare it with the coefficient calculated theoretically, discussing the reason of the difference.

To my mind, the paper is quite well-organized and can be published in the journal. However, there are some remarks that should be taken into account. On page 6 (table 2) the values of the electronic specific heat coefficient differ very much. The most interesting point is connected with the calculated value and measured one. We can see, that the theoretical value is 3 – 5 times higher than the measured one. However, the experimental result for existing magnetic field is much closer to the theoretical one. This effect should be explained more carefully. I should also emphasize that the authors use only one value of the magnetic field, and it does not give full understanding of the influence of magnetic effects. It would be useful to give the results for different magnetic fields, or at least to give some theoretical estimates of dependence between B and \gamma.

Also, there are some minor remarks, which can be corrected before the publication. Firstly, in the reference list there are a lot of papers (about 1/3) which are written by themselves. Of course, if we are compare the numerical results, they are appropriate, but as for the introduction it is better to use more papers of another authors. Also there are some typos: for example, on line 89 the word "Experiment" should not start from capital letter (or it is necessary to replace ";" by "."). On line  9 (page  7) it is necessary to correct the word "Simmltaneously".

Author Response

We are very thankful  for valuable critical remarks, both to Editor and  Referees. We took into account all critical comments. The manuscript  is improved, and it looks better. Below, we respond to critical comments of the  Referees.

Referee 3

Thank you for your kind words about our work, we answered to all critical remarks. The present work contains several new references of  other authors.

- To my mind, the paper is quite well-organized and can be published in the journal. However, there are some remarks that should be taken into account. On page 6 (table 2) the values of the electronic specific heat coefficient differ very much. The most interesting point is connected with the calculated value and measured one. We can see, that the theoretical value is 3 – 5 times higher than the measured one. However, the experimental result for existing magnetic field is much closer to the theoretical one. This effect should be explained more carefully. I should also emphasize that the authors use only one value of the magnetic field, and it does not give full understanding of the influence of magnetic effects. It would be useful to give the results for different magnetic fields, or at least to give some theoretical estimates of dependence between B and \gamma.                                                                                                    Authors: This part of the text is significantly expanded, we explain why γ0 increases with increasing  the magnetic field, we refer to the references, where γ0 vs. B was determined exactly for Sc, Y and Lu compounds. Table 2 is slightly changed and described.

-Also, there are some minor remarks, which can be corrected before the publication. Firstly, in the reference list there are a lot of papers (about 1/3) which are written by themselves. Of course, if we are compare the numerical results, they are appropriate, but as for the introduction it is better to use more papers of another authors. Also there are some typos: for example, on line 89 the word "Experiment" should not start from capital letter (or it is necessary to replace ";" by "."). On line  9 (page  7) it is necessary to correct the word "Simmltaneously".                                                                                        Authors: All these suggestions are considered in the revised paper. We also used more references of another authors. We also added a new text to Introduction (below title of the section) and  some new references.

With regards,

Andrzej  Ślebarski